# Pushing Gradient towards Zero: A Novel Pruning Method for Large Language Models

## Abstract

Recently, large language models (LLMs) have attracted widespread attention due to their dominating performance on some complex language modelling tasks. However, because of their massive size, LLMs require huge amounts of GPU resources in inference which limits their usability. In this paper, we propose an effective pruning method termed PGZ(Pushing Gradient towards Zero), which prunes LLMs in one-shot, without any retraining. The method consists of a new gradual pruning method and a novel weight reconstruction method where gradient is pushed towards zero. More precisely, we construct a loss function based on gradient information and optimize it leveraging second-order information implicitly. In addition, the inherent nature of PGZ makes it suitable for parallelization. Notably, we conduct a thorough evaluation of PGZ on LLaMA-7B,13B,30B,65B across various language benchmarks. Experimental results demonstrate that PGZ consistently outperforms the existing pruning methods for LLMs in unstructured pattern and semi-structured (2:4 and 4:8) pattern. PGZ is also competitive in terms of zero-shot tasks and is compatible with weight quantization approaches.

## 1 INTRODUCTION

Recently, a growing number of Large Language Models (LLMs) demonstrate excellence in a wide range of language tasks. However the computational and storage cost of LLMs makes them difficult to deploy. Taking LLaMA-65b (Touvron et al., 2023) as an example, it has 65 billion parameters and thus even it uses a compact float16 format for inference, its parameters still occupy 120 GB of memory which makes inference on a single A100 GPU infeasible. Model compression (Hoefler et al., 2021; Gholami et al., 2022) is the standard technique to reduce these overheads. There are many pruning techniques suitable for the model with up to a few hundred million parameters. However, these methods are incapable of handling with billion-parameter models. One reason is that these top-performing methods usually require model retraining to reduce loss of accuracy, which is extremely expensive for billion-parameter models. We thus turn to post-training methods, which is also highly challenging to scale to billions of parameters.

**Contribution.** In this paper, we propose PGZ(Pushing Gradient towards Zero), a novel one-shot pruning method which works by reducing the pruning problem to a set of optimization sub-problems which aim to push all modified gradients towards zero. It then solve these sub-problems via a modified gradient descent method. PGZ is efficient enough to execute on models with billions of parameters in at most a few hours. Additionally, we need only a single NVIDIA A100 GPU with 40GB of memory and PGZ is accurate enough and the accuracy loss is negligible.

Our experiments, as shown in Figure, lead to some conclusions. Firstly, PGZ (both unstructured pruning and semi-structured pruning) can be applied to models with billions of parameters, with less accuracy loss than existing one shot pruning methods. Secondly, we show that compared with existing one shot pruning methods, PGZ can provide a more robust result in the joint sparsification and quantization regime, where some components in models are pruned while the rest are quantized to 4 bits.

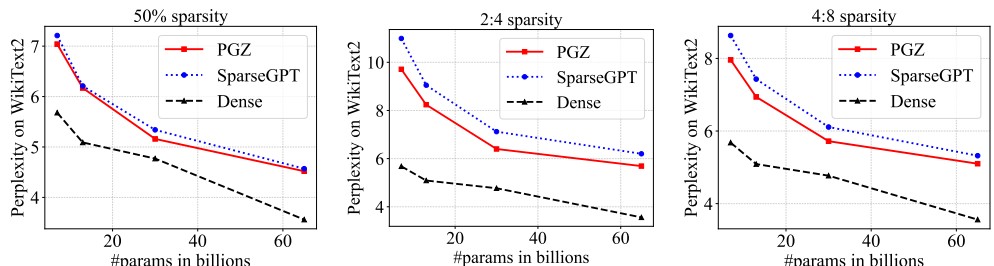

Figure 1: Pruning LLaMA family to different sparsity, comparing PGZ with the FP16 baseline and SparseGPT.

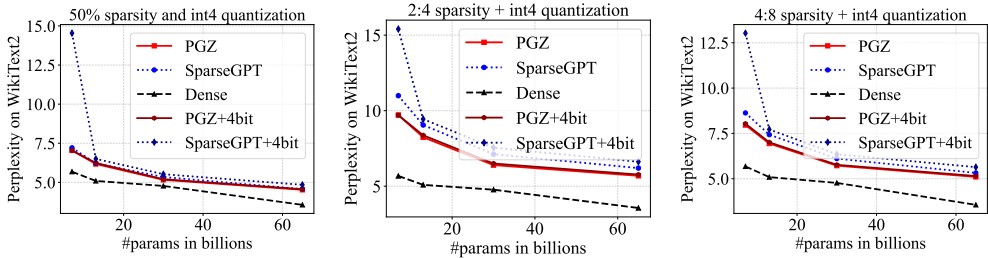

Figure 2: Joint doing pruning and int4 quantization on LLaMA family, comparing PGZ with the FP16 baseline and SparseGPT.

## 2 RELATED WORK

Although pruning methods can date back to the 1980s (Mozer & Smolensky, 1988; Kruschke, 1988), recent advances in deep learning and its potential for applications in embedded systems has led to an increasing number and variety of algorithms for pruning deep neural networks. Pruning methods fall broadly into three categories: unstructured pruning, structured pruning and semi-structured pruning (Zhou et al., 2021; Hubara et al., 2021; Anwar et al., 2017). Unstructured methods remove individual weights at any location, which are fine-grained and can achieve extremely high compression ratio, but are unfriendly to modern hardware. Structured methods remove parameters in groups which are friendly to hardware but limit the expressiveness of the models. Recently semi-structured pruning methods are proposed, which can maintain the advantages of both unstructured pruning methods and structured pruning methods simultaneously on specifically designed GPUs.

**Semi-structured Pruning.** The N:M pruning methods remove N weights out of consecutive M weights. This constraint on sparsification allows for sparse representations similar in flexibility to those of unstructured approaches but also permits efficient hardware implementation as well (Holmes et al., 2021). Zhou et al. (2021) extends STE to tackle the problem of training N:M sparse neural networks. Hubara et al. (2021) introduces a novel transposable fine-grained sparsity mask and formulate the problem of finding the optimal transposable-mask as a minimum-cost flow problem. It achieves a 2× speed-up with no o accuracy degradation. Anwar et al. (2017) introduces structured sparsity at various scales for convolutional neural networks: feature map-wise, kernel-wise, and intra-kernel strided sparsity. This structured sparsity is very advantageous for direct computational resource savings on embedded computers.

**Post-training Pruning.** Post-training compression methods originally are popular in model quantization which contains the AdaRound method (Nagel et al., 2020), BRECQ (Li et al., 2021), OBQ (Frantar & Alistarh, 2022). Frantar et al. (2022) proposes GPTQ, a quantization method for models with billions of parameters. Hubara et al. (2021); Frantar & Alistarh (2022) extends post-training method to model pruning. While these approaches can produce good results for models up to 100 million parameters in a few GPU hours, scaling them to networks orders of magnitude larger is challenging. Recently, some one-shot pruning methods for LLMs are proposed, the most representative of which is SparseGPT (Frantar & Alistarh, 2023) and Wanda (Sun et al., 2023). SparseGPT works by reducing the pruning problem to a set of extremely large-scale instances of sparse regression. It then solves these instances via a new approximate sparse regression solver.

Wanda removes weights with the smallest magnitudes multiplied by the corresponding input activations on a per-output basis.

## 3 BACKGROUND

**Transformer.** LLMs are typically built with transformers (Vaswani et al., 2017). A LLM model consists of an embedding input layer and a number of decoder layers. Each decoder layer comprises a self-attention module. The self-attention module maps input features into a set of queries, keys, and values $\mathbf{q}, \mathbf{k}, \mathbf{v}$, via linear projection layers with weight matrices $\mathbf{W_q}, \mathbf{W_k}, \mathbf{W_v}$. Given $\mathbf{q}, \mathbf{k}, \mathbf{v}$, it computes the outputs o as

$$\mathbf{o} = \text{softmax}(\mathbf{q}\mathbf{k}^T)\mathbf{v} \tag{1}$$

The outputs are then projected by a linear layer with a weight matrix $\mathbf{W_o}$. And MLP layers are followed.

**Layer-Wise Pruning.** Post-training compression is usually done by splitting the full-model compression problem into layer-wise sub-problems, whose solution quality, as shown in formula (2), is measured in terms of the squared error between the output, for given inputs $\mathbf{X}$, of the uncompressed layer with weights $\mathbf{W}$ and that of the compressed one. Thus, the objective is to find a matrix of pruned weights $\widehat{\mathbf{W}}$ which minimizes the squared error defined in (2).

$$F = ||\mathbf{W}\mathbf{X} - \widehat{\mathbf{W}}\mathbf{X}||_F^2 \tag{2}$$

where $||.||_F$ is Frobenius norm, $\mathbf{W}$ is the origin weights, $\widehat{\mathbf{W}}$ is the weights of the pruned layer, $\mathbf{X}$ is the input of the layer. Formula (2) can be restated as

$$\begin{aligned} F &= ||\mathbf{W}\mathbf{X} - (\mathbf{W} + \Delta\mathbf{W})\mathbf{X}||_F^2 \\ &= ||\Delta\mathbf{W}\mathbf{X}||_F^2 \end{aligned} \tag{3}$$

where $\Delta\mathbf{W}$ is the variation of weight during pruning. Formally, minimizing $F$ can be rewrote as

$$\text{argmin}_{\Delta\mathbf{W}}||\Delta\mathbf{W}\mathbf{X}||_F^2. \tag{4}$$

## 4 THE PGZ ALGORITHM

### 4.1 THE MODIFIED GRADIENT

The primary work during pruning is to minimize $||\Delta\mathbf{W}\mathbf{X}||_F^2$. It is easily verified that (4) can be converted into (5).

$$||\Delta\mathbf{W}\mathbf{X}||_F^2 = \sum_{i=1}^{n} ||\mathbf{X}^T[\widehat{\mathbf{W}}_i - \mathbf{W}_i]||_2^2 \tag{5}$$

where $\widehat{\mathbf{W}}_i$ and $\mathbf{W}_i$ is the $i$-th column of $\widehat{\mathbf{W}}^T$ and $\mathbf{W}^T$ respectively, $n$ is the number of columns in $\Delta\mathbf{W}^T$.

$$\min_{\widehat{\mathbf{W}}} ||\Delta\mathbf{W}\mathbf{X}||_F^2 = \sum_{i=1}^{n} \min_{\widehat{\mathbf{W}}_i} ||\mathbf{X}^T[\widehat{\mathbf{W}}_i - \mathbf{W}_i]||_2^2 \tag{6}$$

Two different columns in the transposed weight matrix are independent. This feature help us divide the problem into $n$ independent quadratic optimization problems. These problems can be solved in parallel. We extract one item from (6) and rewrite it as

$$\begin{aligned} w &= \text{argmin}_{\mathbf{w}}||\mathbf{X}^T[\mathbf{w} - \mathbf{W}_i]||_2^2 \\ &= \text{argmin}_{\mathbf{w}}(\mathbf{w}^T(\mathbf{X}\mathbf{X}^T)\mathbf{w} - 2\mathbf{W}_i^T\mathbf{X}\mathbf{X}^T\mathbf{w}) \end{aligned} \tag{7}$$

Due to $\mathbf{X}\mathbf{X}^T$ is positive-definite(If it is not, we add $\lambda I$ to it), (7) is equal to pushing its gradient towards zero.

$$\text{grad}(\mathbf{w}) = 2(\mathbf{X}\mathbf{X}^T)\mathbf{w} - 2\mathbf{X}\mathbf{X}^T\mathbf{W}_i = 0 \tag{8}$$

Some problems still exist when solving equation (8). The first is the numeric overflow problem, which is caused by the huge gradient during weight reconstruction. In addition, $\mathbf{X}\mathbf{X}^T$ calculated from each layer's input usually has a large condition number and thus makes solving (8) a huge challenge. The third is the number of equation (8) is equal to that of columns in the weight matrix and thus it makes traditional methods of solving equations unfeasible due to its inefficiency and it also limits pruning granularity.

To solve these problems, we propose several simple and effective methods including input pre-processing, matrix decomposition. In the rest of the section, we focus on the second method. We denote $\mathbf{H}$ as $\mathbf{X}\mathbf{X}^T$ and leverage Cholesky decomposition $\mathbf{H} = \mathbf{L}^T\mathbf{L}$ to reduce the condition number of weight matrix. Then $\mathrm{grad}(\mathbf{w})$ can be rewrote as

$$
\begin{aligned}
\mathrm{grad}(\mathbf{w}) &= 2\mathbf{H}\mathbf{w} - 2\mathbf{H}\mathbf{W}_i \\
&= 2(\mathbf{L}^T\mathbf{L})\mathbf{w} - 2(\mathbf{L}^T\mathbf{L})\mathbf{W}_i \\
&= \mathbf{L}^T(2\mathbf{L}\mathbf{w} - 2\mathbf{L}\mathbf{W}_i) \\
&= \mathbf{L}^T S(\mathbf{w})
\end{aligned}
\tag{9}
$$

For convenience, $\mathbf{b} \triangleq -2\mathbf{L}\mathbf{W}_i$, $S(\mathbf{w}) \triangleq 2\mathbf{L}\mathbf{w} + \mathbf{b}$. Due to $\mathbf{L}$ is non-singular, pushing $\mathrm{grad}(\mathbf{w})$ towards zero is equivalent to pushing $S(\mathbf{w})$ towards zero.

## 5 PRUNING METHOD

Typically, pruning method comprises mask selection and weight reconstruction. Mask selection shifts $S(\mathbf{w})$ away from zero and weight reconstruction pushes $S(\mathbf{w})$ towards zero. We construct pruning metric and develop weight reconstruction strategy based on $S(\mathbf{w})$. The purpose of this section is to introduce a novel pruning method, which consists of a new pruning metric, a new pruning method and a new weight reconstruction strategy.

### 5.1 PRUNING METRIC

For each individual weight $\mathbf{w}_j$, we evaluate its importance by its score, which is defined as the product of $\Delta S_1$ and $\Delta S_2$ introduced by removing $\mathbf{w}_j$. $\Delta S_1$ and $\Delta S_2$ is stated as formula (10).

$$
\begin{aligned}
\Delta S_1 &= ||S(\mathbf{w}) - S(\mathbf{w}|\mathbf{w}_j = 0)||_2 \\
\Delta S_2 &= |S(\mathbf{w}|\mathbf{w}_j = 0)_j \mathbf{w}_j|
\end{aligned}
\tag{10}
$$

It is easily verified that

$$
\begin{aligned}
\Delta S_1 &= 2||\mathbf{L_j}||_2|\mathbf{w}_j| \\
\Delta S_2 &= |\mathbf{L}_{jj}\mathbf{w}_j^2 - (2\mathbf{L}\mathbf{w} + \mathbf{b})_j\mathbf{w}_j|
\end{aligned}
\tag{11}
$$

The score $\mathbf{S}_j$ for weight $\mathbf{w}_j$ is defined as

$$
\mathbf{S}_j = \Delta S_1 \Delta S_2
\tag{12}
$$

We take the scores calculated at previous iterations into consideration. At $(k+1)$-th iteration, the score $S_j^{k+1}$ is defined as

$$
\mathbf{S}_j^{k+1} = (1-\alpha)\mathbf{S}_j + \alpha\mathbf{S}_j^k
\tag{13}
$$

where $\mathbf{S}_j$ is calculated according to (12), $\mathbf{S}_j^k$ is the score at $k$-th iteration, $\alpha$ is a hyber-parameter, we set $\alpha$ to be 0.1 in the paper.

### 5.2 GRADUAL PRUNING METHOD

In this subsection, we introduce a novel gradual pruning method in which the sparsity is increased from the initial sparsity value $s_0$ at a fixed step size $\Delta s$. $\Delta s$ is also called sparsity step size. At step $t$, the sparsity value is

$$
s_t = s_0 + \Delta s * t
\tag{14}
$$

The mask is updated per step according to the following formula.

$$
\mathbf{mask}_{t+1} = \mathbf{mask}_t \mid \Delta\mathbf{mask}_t
\tag{15}
$$

$$
\mathbf{0} = \mathbf{mask}_t \, \& \, \Delta\mathbf{mask}_t
\tag{16}
$$

| is bitwise OR operator, & is bitwise AND operator.

## 5.3 WEIGHT RECONSTRUCTION

Weight reconstruction aims to push the modified gradient $S(\mathbf{w})$ towards zero, which is modelled as an optimization problem. The new loss function is defined as

$$f(\mathbf{w}) = \alpha||S(\mathbf{w})||_1 + \beta S(\mathbf{w})^T S(\mathbf{w}) \tag{17}$$

where $\alpha$ and $\beta$ is hyper-parameters, $S(\mathbf{w})$ is the modified gradient, $\mathbf{w}$ is weight vector.

The goal of weight reconstruction is to minimize $f(\mathbf{w})$. We leverage Taylor expansion and get the approximation formula

$$f(\mathbf{w} + \Delta\mathbf{w}) \approx f(\mathbf{w}) + \nabla f(\mathbf{w})^T \Delta\mathbf{w} \tag{18}$$

The key is to find out a direction $\Delta\mathbf{w}$ such that $\nabla f(\mathbf{w})^T \Delta\mathbf{w}$ is negative from (18). After mathematical derivation, the gradient of $f(\mathbf{w})$ with respect to $\mathbf{w}$ is

$$\nabla f(\mathbf{w}) = 2\mathbf{L}^{\mathrm{T}}\nabla_S f \tag{19}$$

$-\mathbf{L}^{\mathrm{T}}\nabla_S f$ is chosen to be the descent direction. Thus each iteration is depicted as

$$\begin{aligned} w_{t+1} &= w_t - \lambda\mathbf{L}^{\mathrm{T}}\nabla_S f \\ &= w_t - \lambda\mathbf{L}^{\mathrm{T}}(\alpha\mathrm{sign}(S) + 2\beta S) \end{aligned} \tag{20}$$

where $t$ is the optimizing step, $\lambda$ is the learning rate. We learn from Lion optimizer (Chen et al., 2023) and propose a modified gradient descent method. The method converges to high precision solution rapidly. In addition, it can be extended to parallel scenarios, which speeds up pruning.

Finally, we present the full pseudocode for the modified gradient descent method.

---

**Algorithm 1** Modified Lion Optimizer, $\beta_1 = 0.9$, $\beta_2 = 0.99$, $\rho = 1.4$, $\mu = 0.01$, $\lambda = 8e - 4$

---

    **for** k=0,1,2,... **do**
        $g \leftarrow \nabla f$
        $c \leftarrow \beta_1 m + (1 - \beta_1)\,g$
        $m \leftarrow \beta_2 m + (1 - \beta_2)\,g$
        $v \leftarrow \rho\,\mathrm{sign}\,(c) + (2 - \rho)\,g$
        $\theta \leftarrow (1 - \mu\lambda)\,\theta - \lambda\frac{v}{k+1}$
    **end for**

---

## 5.4 JOINT SPARSIFICATION QUANTIZATION

The algorithm 1 indicates that pruning and quantization can be merged into a single compression procedure in which we do pruning and quantization alternately. Quantization is inserted between mask selection and weight reconstruction and we deduce from this that later pruning strategy is influence by earlier quantization, which is different from the prior technique (Frantar & Alistarh, 2022). Due to pruning and quantization is excuted simultaneously in a single pass, it has no more costs than PGZ.

# 6 EXPERIMENTS

## 6.1 SETUP

**Models, Datasets and Evaluation** We primarily evaluate PGZ on the LLaMA model family including LLaMA-7B/13B/30B/65B. All pruning experiments are conducted on a single A100 GPU with 40G of memory. Similar to SparseGPT, we prune Transformer layers sequentially in order, which can significantly reduce memory requirements. Our calibration set consists of 320 sequences (2048 tokens each) sampled from the first shard of the C4 (Raffel et al., 2020) training data according to the observation of increasing the amount of calibration data improves the performance of our pruned models . We measure the performance of the pruned networks via perplexity computed on WikiText-2 (Merity et al., 2016) validation set. We also provide the perplexity metric on other validation sets in Appendix C. As a supplement to perplexity evaluations, We evaluate zero-shot ability of the pruned

models with GPTQ's (Frantar et al., 2022) implementation, which is based on the public evaluation benchmark EleutherAI LM Harness. For zero-shot performance, we evaluate on six common sense benchmarks: BoolQ (Clark et al., 2019), RTE (Wang et al., 2018), WinoGrande (Sakaguchi et al., 2021), ARC Easy and Challenge (Boratko et al., 2018), and OpenbookQA (Mihaylov et al., 2018). We report the accuracy on each benchmark as well as the overall mean accuracy.

**Baselines.** We compare PGZ against SparseGPT which is a novel pruning method applied to LLMs. It scales existing second-order based approaches (Frantar & Alistarh, 2022) to LLMs. We use the implementation of SparseGPT provided by IST-DASLab to reproduce the pruning method in SparseGPT.

## 6.2 SPARSITY PATTERN.

For all pruning methods, we follow the setup of SparseGPT, where a uniform sparsity is imposed for all layers and there is no subsequent retraining. We skip the first embedding layer and the final classification head, as is common in pruning Transformers. Our primary approach to induce sparsity is through unstructured pruning. Considering the potential need for practical speedup, we also conduct evaluations on structured N:M sparsity (Zhou et al., 2021) (Hubara et al., 2021). Specifically, we provide comparisons on 4:8 and 2:4 sparsity patterns.

## 6.3 PREPROCESSING OPERATION

In experiments, we find that there are some outliers (Dettmers et al., 2022) in input which causes the numerical stability issues. To address the problem, we modify the input according to the following formula.

$$\mathbf{X} = \frac{\mathbf{X} - \text{mean}(\mathbf{X})}{\sqrt{\text{mean}(\mathbf{X}\mathbf{X}^T) + \text{eps}}} \tag{21}$$

## 6.4 LANGUAGE MODELING

**Unstructured Sparsity.** For each of the LLaMA models, we adopt different approaches to prune it to unstructured 50% sparsity. we evaluate two methods (SparseGPT, PGZ) on several validation datasets including WikiText2, PTB and C4. Results on WikiText2 are reported in Table 1 and results on other datasets can be found in Appendix. Experimental results illustrate that PGZ outperforms the existing pruning approaches.

**Structured N:M Sparsity.** We now turn our eyes to structured N:M sparsity. Results for structured 4:8 and 2:4 sparsity are shown in the lower parts of Table 1. We can see that PGZ can be easily generalized to structured N:M sparsity. Across 4:8 and 2:4 sparsity, PGZ consistently outperforms baseline approaches.

Table 1: LLaMA perplexity results on WikiText2.

| Models | Sparsity | Dense | SparseGPT | PGZ |
|---|---|---|---|---|
| LLaMA-7b | 50% | 5.68 | 7.23 | **7.04** |
| LLaMA-13b | 50% | 5.09 | 6.21 | **6.17** |
| LLaMA-30b | 50% | 4.77 | 5.31 | **5.16** |
| LLaMA-65b | 50% | 3.56 | 4.57 | **4.52** |
| LLaMA-7b | 2:4 | 5.68 | 11.00 | **9.71** |
| LLaMA-13b | 2:4 | 5.09 | 9.11 | **8.24** |
| LLaMA-30b | 2:4 | 4.77 | 7.16 | **6.40** |
| LLaMA-65b | 2:4 | 3.56 | 6.28 | **5.69** |
| LLaMA-7b | 4:8 | 5.68 | 8.60 | **7.96** |
| LLaMA-13b | 4:8 | 5.09 | 7.40 | **6.94** |
| LLaMA-30b | 4:8 | 4.77 | 6.17 | **5.72** |
| LLaMA-65b | 4:8 | 3.56 | 5.38 | **5.10** |

## 6.5 ZERO-SHOT EXPERIMENTS

To complement the perplexity evaluations, we employ Language Model Evaluation Harness to conduct test on more datasets, including BoolQ, RTE, WinoGrande, ARC-e, ARC-c and OBQA. In these experiments, we set sparsity step size $\Delta s$ 0.1. Results are summarized in Table 2. Averaging the accuracy over the 6 tasks under consideration, our method is competitive with SparseGPT.

Table 2: Accuracies (%) for 6 zero-shot tasks with unstructured 50% sparsity.

| Models | Methods | BoolQ | RTE | WinoGrande | ARC-e | ARC-c | OBQA | Mean |
|---|---|---|---|---|---|---|---|---|
| LLaMA-7B | Dense | 71.7 | 53.4 | 68.0 | 67.7 | 38.6 | 28.0 | 55.1 |
| | SparseGPT | 71.5 | 56.8 | 65.8 | 64.3 | 36.0 | 25.0 | 53.2 |
| | PGZ | **72.3** | **62.8** | 63.9 | 63.1 | **36.7** | 24.2 | **53.8** |
| LLaMA-13B | Dense | 68.3 | 65.3 | 70.0 | 73.6 | 44.0 | 30.6 | 58.9 |
| | SparseGPT | 66.7 | 52.0 | 70.9 | 66.9 | 39.2 | 26.0 | 53.8 |
| | PGZ | 66.6 | 49.5 | **71.0** | **69.4** | **41.7** | **28.8** | **54.6** |
| LLaMA-30B | Dense | 66.9 | 61.4 | 72.4 | 75.3 | 46.9 | 29.4 | 59.6 |
| | SparseGPT | 71.0 | 61.4 | 72.0 | 73.9 | 46.0 | 31.2 | 59.2 |
| | PGZ | **71.1** | 61.0 | 70.8 | **75.2** | **46.5** | **31.6** | **59.4** |
| LLaMA-65B | Dense | 81.8 | 71.8 | 76.9 | 75.4 | 47.2 | 36.4 | 65.0 |
| | SparseGPT | 81.2 | 70.4 | 74.1 | 74.8 | 44.7 | 32.2 | 62.8 |
| | PGZ | **83.0** | 70.0 | **76.7** | 73.9 | **47.1** | **33.0** | **64.0** |

## 6.6 JOINT SPARSIFICATION & QUANTIZATION

The combination of pruning and quantization is another attractive research direction, which can have the advantages of both approaches: computational speedups from sparsity and memory savings from quantization. Specifically, if we compress one model to 50% sparsity + 4-bit weight, and use a bitmask to indicate their positions, then this has the same overall memory consumption as 3-bit quantization. Hence, in table 6, we compare PGZ 50% + 4-bit with state-of-the-art GPTQ 3-bit numbers. It can be seen that 50% + 4-bit models are more accurate than their respective 3-bit versions for LLAMA-7B/13B/30B/65B. We also tested 2:4 and 4:8 in combination with 4-bit on LLAMA.

Table 3: Comparing joint PGZ + 4-bit quantization with 3bit GPTQ on WikiText2.

| Models | sparsity pattern | Dense | 3bit GPTQ | PGZ + 4-bit |
|---|---|---|---|---|
| LLaMA-7b | 50% | 5.68 | 8.07 | **7.04** |
| LLaMA-13b | 50% | 5.09 | 6.63 | **6.24** |
| LLaMA-30b | 50% | 4.77 | 5.69 | **5.20** |
| LLaMA-65b | 50% | 3.56 | 5.04 | **4.56** |
| LLaMA-7b | 2:4 | 5.68 | - | **9.70** |
| LLaMA-13b | 2:4 | 5.09 | - | **8.37** |
| LLaMA-30b | 2:4 | 4.77 | - | **6.50** |
| LLaMA-65b | 2:4 | 3.56 | - | **5.77** |
| LLaMA-7b | 4:8 | 5.68 | - | **8.04** |
| LLaMA-13b | 4:8 | 5.09 | - | **7.01** |
| LLaMA-30b | 4:8 | 4.77 | - | **5.77** |
| LLaMA-65b | 4:8 | 3.56 | - | **5.15** |

## 6.7 ABLATION STUDY

In this section, we will answer the following questions about PGZ:

- What is the relationship between sparsity step size and perplexity?
- What is the relationship between sparsity and perplexity?
- what is the relationship between the number of calibration data and the performance of these pruned models?

**Varying Sparsity Step Size.** We conduct experiments with varying sparsity step size $\Delta s$ for unstructured pruning at different sparsity level, the result of which are depicted in Figure 3. The sparsity is set to 50

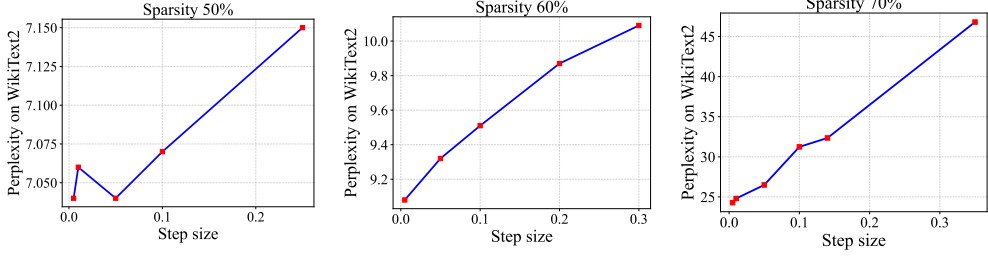

Figure 3: Sparsity step size-vs-perplexity at 50%, 60%, 70% sparsity. The reduction of sparsity step size can improve the performance of the pruned models.

**Varying Sparsity.** We conduct experiments with varying sparsity for unstructured pruning, the results of which are depicted in Figure 4. It can be seen that PGZ and SparseGPT shows similar trends of perplexity increase as the sparsity level gets higher. However, SparseGPT displays a more severe degradation trend.

**The number of calibration samples.** Figure 4 demonstrates that increasing the number of calibration samples improves the model's accuracy performance.

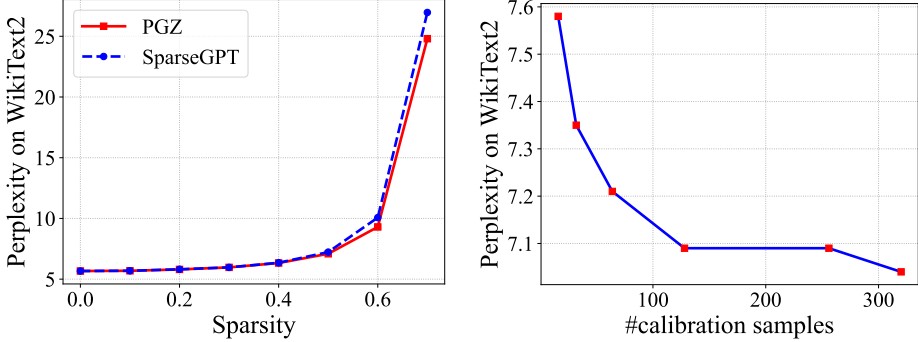

Figure 4: The left shows that the relationship between sparsity and perplexity. The right shows that the relationship between calibration samples and perplexity.

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

# A   APPENDIX

## A.1   ADDITIONAL LANGUAGE GENERATION RESULTS

Tables 6, 7, 8 and 9 show additional results for language generation tasks.

Table 4: Comparing PGZ with SparseGPT on PTB.

| Models | Sparsity | SparseGPT | PGZ |
|---|---|---|---|
| LLaMA-7b | 50% | 78.16 | **56.13** |
| LLaMA-13b | 50% | 37.24 | **35.35** |
| LLaMA-30b | 50% | 26.23 | **25.72** |
| LLaMA-65b | 50% | 28.21 | **27.94** |
| LLaMA-7b | 2:4 | 151.07 | **102.22** |
| LLaMA-13b | 2:4 | 70.06 | **55.36** |
| LLaMA-30b | 2:4 | 32.12 | **31.04** |
| LLaMA-65b | 2:4 | 33.14 | **32.63** |
| LLaMA-7b | 4:8 | 108.29 | **75.21** |
| LLaMA-13b | 4:8 | 47.12 | **41.58** |
| LLaMA-30b | 4:8 | 29.27 | **27.71** |
| LLaMA-65b | 4:8 | **31.07** | 31.55 |

Table 5: Comparing PGZ with SparseGPT on C4.

| Models | Sparsity | SparseGPT | PGZ |
|---|---|---|---|
| LLaMA-7b | 50% | 9.27 | **9.15** |
| LLaMA-13b | 50% | 8.12 | **8.09** |
| LLaMA-30b | 50% | 7.34 | **7.19** |
| LLaMA-65b | 50% | 6.66 | **6.57** |
| LLaMA-7b | 2:4 | 13.73 | **12.11** |
| LLaMA-13b | 2:4 | 11.29 | **10.30** |
| LLaMA-30b | 2:4 | 9.42 | **8.71** |
| LLaMA-65b | 2:4 | 8.38 | **7.80** |
| LLaMA-7b | 4:8 | 10.98 | **10.32** |
| LLaMA-13b | 4:8 | 9.40 | **8.95** |
| LLaMA-30b | 4:8 | 8.20 | **7.83** |
| LLaMA-65b | 4:8 | 7.41 | **7.14** |

Table 6: Comparing joint PGZ + 4-bit quantization with SparseGPT + 4-bit quantization on PTB.

| Models | Sparsity | SparseGPT + 4bit | PGZ + 4bit |
|--------|----------|------------------|------------|
| LLaMA-7b | 50% | 103.02 | **55.53** |
| LLaMA-13b | 50% | 39.94 | **36.63** |
| LLaMA-30b | 50% | 27.29 | **25.86** |
| LLaMA-65b | 50% | 27.62 | **26.69** |
| LLaMA-7b | 2:4 | 210.51 | **102.23** |
| LLaMA-13b | 2:4 | 65.66 | **55.89** |
| LLaMA-30b | 2:4 | 36.19 | **32.16** |
| LLaMA-65b | 2:4 | 37.49 | **34.07** |
| LLaMA-7b | 4:8 | 132.74 | **74.96** |
| LLaMA-13b | 4:8 | 46.47 | **41.83** |
| LLaMA-30b | 4:8 | 30.64 | **28.34** |
| LLaMA-65b | 4:8 | 34.16 | 31.76 |

Table 7: Comparing joint PGZ + 4-bit quantization with SparseGPT + 4-bit quantization on C4.

| Models | Sparsity | SparseGPT + 4bit | PGZ + 4bit |
|--------|----------|------------------|------------|
| LLaMA-7b | 50% | 11.37 | **9.16** |
| LLaMA-13b | 50% | 8.43 | **8.20** |
| LLaMA-30b | 50% | 7.57 | **7.26** |
| LLaMA-65b | 50% | 6.92 | **6.62** |
| LLaMA-7b | 2:4 | 18.17 | **12.11** |
| LLaMA-13b | 2:4 | 11.72 | **10.47** |
| LLaMA-30b | 2:4 | 9.74 | **8.80** |
| LLaMA-65b | 2:4 | 8.83 | **7.87** |
| LLaMA-7b | 4:8 | 13.48 | **10.35** |
| LLaMA-13b | 4:8 | 9.71 | **9.07** |
| LLaMA-30b | 4:8 | 8.45 | **7.89** |
| LLaMA-65b | 4:8 | 7.75 | **7.20** |

