# OpenReview forum: "Pushing Gradient towards Zero: A Novel Pruning Method for Large Language Models"
_ICLR.cc/2024/Conference — Submitted to ICLR 2024_

### Official Review · Reviewer_ZRAv · 2023-11-01

**Soundness:** 3 good
**Presentation:** 2 fair
**Contribution:** 3 good
**Rating:** 5
**Confidence:** 4

**Summary:**

The paper proposes a new one-shot pruning method called PGZ that prunes LLMs without any retraining. PGZ constructs a loss function based on gradient information and optimizes it using second-order information implicitly to push the gradient to zero. Extensive experiments show that the performance of the pruning method outperforms existing pruning method such as SparseGPT on LLaMA series.

**Strengths:**

- The paper is well-written and easy to comprehend.
- It investigates using both pruning and quantization together for exhaustive search to demonstrate that PGZ with 4-bit quantization outperforms 3-bit post-training quantization.

**Weaknesses:**

- While the paper focuses on unstructured and semi-structured pruning, it would be beneficial to discuss relevant structured pruning techniques as well and how semi-structured pruning has advantages and disadvantages compared to those approaches. I encourage the authors to have a broader discussion surrounding pruning-based compression methods.
- Though the proposed method is well and clearly written. I find it hard to form an intuitive understanding of why it could outperform methods like SparseGPT, which also uses second-order information for pruning and weight recovery. To have a high level discussions about the it’s underlying difference would be very helpful.
- Comparison to stronger baselines such as wandb is lacking. Since wandb is a more efficient method (does not require gradients or second order information) for semi-structured pruning and outperforms SparseGPT, the authors should compare their method to wandb in terms of both performance and efficiency. This would lead to better understanding the advantages of the proposed approach.

**Questions:**

- Since the tasks evaluated are mostly multiple-choice questions or classification questions, it's hard to see how one-shot pruning will affect generation abilities of the model. Do you think such pruning will hurt model generation? Would you consider expanding evaluations to more long-form generations?
- How can I keep finetuning a model that has a semi-structurally sparse? Will the same technique be applied to instruction tuned models like llama-2-chat?

---

### Official Review · Reviewer_DWaW · 2023-11-02

**Soundness:** 3 good
**Presentation:** 3 good
**Contribution:** 2 fair
**Rating:** 5
**Confidence:** 3

**Summary:**

This paper presents a novel pruning method called PGZ for large language models (LLMs) that does not require retraining and is compatible with weight quantization approaches. The authors show that PGZ consistently outperforms existing pruning methods for LLMs in terms of performance. The paper also includes experimental results and analysis to support the effectiveness of PGZ. Overall, the paper's contribution is a simple and effective pruning approach for LLMs that can be parallelized and used for zero-shot tasks.

**Strengths:**

Originality: The paper's contribution is a novel pruning method for LLMs that does not require retraining and is compatible with weight quantization approaches. The method consists of a new gradual pruning method and a novel weight reconstruction method where gradient is pushed towards zero.

Quality: The paper is of high quality, with a thorough evaluation of PGZ on LLMs across various language benchmarks. The authors provide experimental results and analysis to support the effectiveness of PGZ.

Clarity: The paper is well-written and well-organized, making it easy to follow and understand.

Significance: The paper's significance lies in its potential to improve the efficiency and effectiveness of LLMs.

**Weaknesses:**

1. An important baseline: Mingjie Sun, Zhuang Liu, Anna Bair, J. Zico Kolter  A Simple and Effective Pruning Approach for Large Language Models, first appearing in June on arXiv, is not compared in the experiments. The authors may need to explain that.


2. While the paper presents PGZ as an effective pruning method for LLMs, it does not discuss the limitations of the proposed method. For example, the authors could discuss the impact of pruning on model interpretability, the effect of pruning on model robustness, and the scalability of the proposed method to larger LLMs.

3. Insufficient discussion on the computational cost and GPU/CPU speedup of the proposed method: While the paper briefly mentions that PGZ can be parallelized and used for zero-shot tasks, it does not provide a detailed analysis of the computational cost of the proposed method and inference speedup of the model after pruning. Specifically, the authors could provide more insights into the training time, memory requirements, and other computational costs associated with PGZ and also for the model after pruning.

**Questions:**

see weaknesses

---

### Official Review · Reviewer_NTSu · 2023-11-03

**Soundness:** 3 good
**Presentation:** 2 fair
**Contribution:** 2 fair
**Rating:** 5
**Confidence:** 4

**Summary:**

The paper proposes a novel pruning method called PGZ for large language models (LLMs) that can prune the models without any retraining. The method consists of a new gradual pruning method and a novel weight reconstruction method where the gradient is pushed towards zero. The authors construct a loss function based on gradient information and optimize it leveraging second-order information implicitly. The inherent nature of PGZ makes it suitable for parallelization. The paper evaluates PGZ on LLaMA-7B, 13B, 30B, 65B across various language benchmarks. Experimental results demonstrate that PGZ consistently outperforms the existing pruning methods for LLMs in unstructured pattern and semi-structured (2:4 and 4:8) pattern. PGZ is also competitive in terms of zero-shot tasks and is compatible with weight quantization approaches.

**Strengths:**

- PGZ can prune large language models without any retraining, making it suitable for parallelization and compatible with weight quantization approaches.
- Experimental results demonstrate the performance of PGZ in unstructured pattern and semi-structured (2:4 and 4:8) pattern.

**Weaknesses:**

- The paper does not provide a comparison of the computational cost of PGZ with existing pruning methods.
- The performance of PGZ shows marginal improvements compared to existing methods like SparseGPT.

**Questions:**

what's the advantages of PGZ compared to other pruning methods like SparseGPT?

---

### Official Review · Reviewer_ERPv · 2023-11-05

**Soundness:** 2 fair
**Presentation:** 1 poor
**Contribution:** 3 good
**Rating:** 3
**Confidence:** 3

**Summary:**

The authors propose PGZ a pruning method for Large Language models that aims to reduce loss during pruning by minimizing the gradient of the weight reconstruction loss. The method leverages input preprocessing and matrix decomposition to stabilize numerical issues and inefficiency. The authors conduct experiments with LLAMA generative language models on the WikiText-2 text dataset and also evaluate zero-shot performance on 6 additional benchmarks.

**Strengths:**

1. The proposed method shows lower perplexity than SparseGPT on WikiText-2 for all evaluated LLaMA models
2. The proposed method shows higher mean performance across the six zero-shot LM Evaluation Harness tasks than SparseGPT

**Weaknesses:**

1. Overall writing can be improved substantially, the paper has numerous typos, spelling errors and missing experimental details that make assessment of the technical contribution difficult.
   - Example: In Contributions, "PGZ is efficient enough to execute on models with billions of parameters in at most a few hours" -- execute what: pruning of the model? inference on a dataset?
   - Example: Missing figure reference in line 1 of Paragraph 3 on Page 1
   - Example: What is the value of epsilon in Sec 6.3
   - Example: Missing citations for PTB dataset in Appendix
   - Nit: Bolding in Table 2 is distracting and inconsistent; the Dense baseline is ignored when it has the best performance
2. Limited evaluations and comparisons with other pruning baselines such as AdaPrune and magnitude pruning.
3. One of the primary benefits of pruning, especially for structured pruning, is to reduce computation requirements such that inference latency is reduced. The paper performs no experiments examining potential speedups.

I recommend another round of revisions before submission.

**Questions:**

1. In Figure 1, the comparisons between the pruned models (PGZ, Sparse GPT) and the baseline models are matched to a set number of parameters. If the pruned models have parameters removed, would it be the case that they would be shifted along the x-axis (parameters)?
2. How does inference latency with PGZ compare with SparseGPT and the dense baseline network?
3. In Contributions, it is suggested that PGZ can prune the target model in "a few hours". In Figure 4, the performance across various number of calibration examples are shown. What are the actual computational and wall-clock requirements to calibrate and prune each model?

---

### Meta-Review · Area_Chair_9kc6 · 2023-12-02

**Metareview:**

The paper proposes a new one-shot pruning method called PGZ that prunes LLMs without any retraining. It received scores of 3555. All the authors had some concerns about the paper: (1) overall writing can be improved substantially, (2) limited evaluations and comparisons with other pruning baselines such as magnitude pruning, AdaPrune and Wanda, (3) the paper performs no experiments examining potential speedups, (4) lack of experimental results that go beyond multiple-choice questions or classification questions, such as long-form generations for instruction-tuned models. No rebuttal is provided. Therefore, the AC would like to recommend rejection of the paper.

**Justification For Why Not Higher Score:**

All the authors gave reject recommendations, and no rebuttal is provided.

**Justification For Why Not Lower Score:**

N/A

---

### Decision · Program_Chairs · 2024-01-16

Reject